# Experiences, barriers, and facilitators of health data use among performance monitoring teams (PMT) of health facilities in Eastern Ethiopia: A qualitative study

Admas Abera[1]*, Abebe Tolera[1], Biruk Shalmeno Tusa[1], Adisu Birhanu Weldesenbet[1], Assefa Tola[1], Tilahun Shiferaw[2], Alemayehu Girma[3], Rania Mohammed[4], Yadeta Dessie[1]

1 School of Public Health, College of Health and Medical Sciences, Haramaya University, Harar, Ethiopia, 2 Department of Information Sciences, College of Computing and Informatics, Haramaya University, Haramaya, Ethiopia, 3 Policy and Plan Directorate, Dire Dawa Administration Health Bureau, Dire Dawa, Ethiopia, 4 Policy and Plan Directorate, Harari Regional Health Bureau, Harar, Ethiopia

* admasabera10@gmail.com

**Data Availability Statement:** All relevant data are within the paper.

## Abstract

### Background

Routine health data is crucial in decision-making and improved health outcomes. Despite the significant investments in improving Ethiopia's Performance Monitoring Team (PMT), there is limited evidence on the involvement, implementation strategies, and facilitators and barriers to data utilization by these teams responding to present and emerging health challenges. Therefore, this study aimed to explore the PMT experiences, facilitators, and barriers to information use in healthcare facilities in Eastern Ethiopia.

### Method

This study employed a phenomenological study design using the Consolidated Framework for Implementation Research (CFIR) to identify the most relevant constructs, aiming to describe the data use approaches at six facilities in Dire Dawa and Harari regions in July 2021. Key informant interviews were conducted among 18 purposively selected experts using a semi-structured interview guide. Thematic coding analysis was applied using a partially deductive approach informed by previous studies and an inductive technique with the creation of new emerging themes. Data were analyzed thematically using ATLAS.ti.

### Results

Study participants felt the primary function of PMT was improving health service delivery. This study also revealed that data quality, performance, service quality, and improvement strategies were among the major focus areas of the PMT. Data use by the PMT was affected by poor data quality, absence of accountability, and lack of recognition for outstanding performance. In addition, the engagement of PMT members on multiple committees negatively impacted data use leading to inadequate follow-up of PMT activities, weariness, and insufficient time to complete responsibilities.

**Funding:** This work would not be possible without the financial support of the Doris Duke Charitable Foundation (DDCF) under grant number 2017187. The mission of the Doris Duke Charitable Foundation is to improve the quality of people's lives through grants supporting the performing arts, environmental conservation, medical research, and child well-being, and through the preservation of the cultural and environmental legacy of Doris Duke's properties. The funders had no role in study design, data collection and analysis, decision to publish, or preparation of the manuscript. The funder website can be found here: https://www.ddcf.org/.

**Competing interests:** The authors have declared that no competing interests exist.

## Conclusion

Performance monitoring teams in the health facilities were established and functioning according to the national standard. However, barriers to operative data use included PMT engagement with multiple committees, poor data quality, lack of accountability, and poor documentation practices. Addressing the potential barriers by leveraging the PMT and existing structures have the potential to improve data use and health service performance.

## Introduction

Health systems are complex and continually changing to adapt to political, economic, social, technological, and epidemiological realities within constrained resources, particularly in low- and middle-income countries (LMICs) [1–3]. Health facilities must cope with these changing realities through organized management and leadership, which require reliable data for developing a comprehensive policy package for health sector reforms [2, 4].

There has been an increased need for health service performance strengthening to manage population needs through effective leadership [5–7], and an improved health facility data use culture [8–10]. The increasing demand and capacity to use data appear more critical than the expanding supply of evidence [3, 11] for improved access and quality of care [4, 12]. Collection, processing, transforming, communicating, and using service delivery reports and administrative records are crucial in decision-making for improved health outcomes [13, 14].

Data use is the process through which decision-makers and stakeholders explicitly evaluate information in one or more steps of policymaking, program planning, management, or service provision [6]. In the Ethiopian context, the PMT is a multidisciplinary health workforce team primarily responsible for data use [15, 16]. By effectively using the data, targeted health service delivery improvements meets the population's needs [17].

However, in many developing countries, the quality of the evidence needed to generate valid information to make decisions about health programs could be better or more [18–20], mainly due to an ineffective data use culture [20]. The major bottlenecks for information use include inadequate infrastructure, leadership turnover, dysfunctional external relations [21], deficient data collection, and limited resources [13]. In addition, a shortage of skilled workforce, an imbalance in skills, geographic misdistribution, difficulty in inter-professional collaboration, inefficient use of resources, and burnout were found to affect the health service quality [22].

To date, most efforts to strengthen the health information systems have primarily focused on digitization, improving data quality and analysis, and identifying problems; however, the ultimate goal is utilizing information to problem solve, which requires time to build an information use culture [8]. Therefore, engagement (including involvement, commitment, effort or observable behaviour, a positive effect, or some combination of these) of healthcare leaders and managers is pivotal if we are to improve the Health Information System (HIS) and hence, the health service delivery [4, 11].

The engagement level and capacities of health managers and performance monitoring teams to respond to current and emerging issues still need to be better understood [22]. Without radical structural and systemic changes, the existing governance structures and management systems will continue to fail to address the gaps in health service delivery [2]. Furthermore, despite significant investments in performance monitoring teams in Ethiopia, a joint performance renewal effort must be improved. There is also paucity of studies addressing

the facilitators, and challenges influencing health data use to improve primary health care delivery in LMICs. Therefore, this study aimed to identify the PMT experiences, facilitators and barriers to information use in the healthcare facilities in the Harari region and Dire Dawa city administration in Eastern Ethiopia.

# Methods and materials

## Study setting

The study was conducted in selected Dire Dawa and Harari public health facilities. Dire Dawa is located 515 kilometers from Addis Ababa and has an estimated population of 341,834, with 68.23% living in urban areas [23]. Data from the public health facilities indicate 15 health centers (8 urban and seven rural), two hospitals, and 32 health posts under the city administration. Harar is the capital of Harari, one of the regional states of Ethiopia, located 517 Km east of Addis Ababa and 48 km south of Dire Dawa. According to the 2007 census, the region has a total population of 183,415 people [23], and more than half (54.18%) reside in urban areas [24]. The region has three government hospitals, one university teaching hospital, two private hospitals, nine health centers, and 24 health posts.

## Study design and participants

This study used a phenomenological study design. The Consolidated Framework for Implementation Research (CFIR) was used to identify the most relevant constructs to describe data use approaches. The CFIR framework is an evidence-based framework from multiple disciplines providing a comprehensive arrangement of paradigms that influence complex implementations [25]. The framework was adapted through a qualitative theme reduction process and has five significant domains (inner setting, outer setting, intervention characteristics, individuals involved, and implementation process) with associated components. From these domains, we chose to focus specifically on the domains of inner settings, outer settings and individuals involved based upon their relevance to our research question, which is to identify the different factors influencing the data use practice of the PMT. The three domains used in this study were: 1) Inner setting of data use practice (HIS input & infrastructure, communication between PMT and other health workers, PMTs competing priorities, engagement of PMT, staff turnover, and PMT structure at health facilities); 2) Outer settings (policy and guidelines, accountability mechanism, supervision and mentorship, and recognition); and 3) Individuals involved (value for data, workload, readiness and perception). The framework was applied in the design and during data collection phase of this study.

   Two hospitals and four health centers were randomly selected from a list of facilities in the two regions. A purposive sampling technique was employed to select the key informants. Eighteen interviews were conducted until data saturation was achieved. The key informants were PMT members from the health facilities, including hospital medical directors, heads of health centers, heads of departments (outpatient, inpatient, emergency, maternal and child health, pharmacy, laboratory, nursing, human resource, and finance team leads), and the Health Management Information System (HMIS) officers.

## Data collection

We conducted key informant interviews using a semi-structured interview guide. The guide was comprised of questions related to the respondents' socio-demographic characteristics, the process of the PMT establishment and implementation strategy process, data use processes

and practices, barriers to data use, motivating factors for data use, and recommended mechanisms to improve the data use practice.

## Data quality control

Data were collected between July 05, 2021, and July 24, 2021, by interviewers fluent in local languages (Amharic and Afan Oromo) and English, with graduate-level education and experience in qualitative data collection. All data collection team members underwent a one-day training workshop to understand the interview guide and study objectives. The data collection activities were supervised daily by the study investigators. All interviews were conducted privately and lasted 20–60 minutes.

## Data processing and analysis

The interviews were transcribed verbatim, and notes were taken in the field. The interviews were translated into English and translated back to the original languages to confirm the accuracy. A codebook was developed based on the initial review of the transcripts, and transcripts were systematically coded using ATLAS.ti software. Double coding was initially used, with disagreements resolved by discussion; updates to the code definitions were made when needed. Double coding continued until no new disagreements were identified. Subsequently, summaries for each transcript were written under each code using a matrix. Thematic data analysis described and compared general statements as relationships, themes, and sub-themes emerged in the data. Accordingly, findings were categorized into five themes. Under each theme, sub-themes were defined with verbatim quotes representing opinions to substantiate the results. Researchers read all interview transcripts, counterchecked the transcripts, coded the data, and agreed on the emerging themes and sub-themes. Additionally, research rigor was enhanced through regular discussions between researchers.

## Key themes and sub-themes

In this study, three major themes, five key sub-themes and 16 sub-themes emerged. The main themes and sub-themes are presented in Table 1.

## Ethical considerations

Ethical clearance was obtained from Haramaya University, College of Health and Medical Sciences, Institutional Health Research Ethics Review Committee (IHRERC) (reference number IHRERC/196/2020). Permission was sought from Dire Dawa Administration and the Harari Regional Health Bureaus, and the studied health facilities. In addition, informed written consent was obtained from the study participants before data collection. Furthermore, participants were assured of the confidentiality of the information and their right to withdraw from the study at any time during the study. This article omitted personal identifiers to maintain confidentiality, while neutral identifiers and participants' age were mentioned in direct quotes.

## Results

### Socio-demographic characteristics

Of the total eighteen in-depth interviews conducted, eight were females, and three were health facility heads with a median age (IQR) of 31 (7) years (Table 2).

**Table 1. Categories of key themes and sub-themes emerged.**

| Major Themes | Sub-Themes | |
|---|---|---|
| **Experiences of PMT** | Membership and roles of PMT members at the health facilities | • *Roles and responsibilities of PMT members*<br>• *The current PMT status and experiences* |
| | Relevance of PMT strategies and its implementation | • *Perceived benefits of PMT*<br>• *The modality of PMT meetings*<br>• *Adequacy and convenience of time of PMT meeting*<br>• *The focuses of PMT meetings* |
| **Barriers to data use** | Barriers to data use in the facility | • *Organizational barriers*<br>• *Healthcare workers' work ethic and behavioral barriers*<br>• *Demotivating factors COVID-19 effect on data use* |
| **Facilitators of data use** | Motivation mechanisms to enhancing data use practices in place for PMT members | • *Capacity building practices for PMT members*<br>• *Recognition and non-monetary incentives* |
| | Recommended mechanisms to improve the data use practices | • *Performance-based recognition*<br>• *Continuous supply of resources, and improving HIS infrastructures*<br>• *Strengthening the overall HIS*<br>• *Enhancing the quality of routine data* |

## 1. Experiences of PMT

**Membership and roles of PMT at the health facilities.** Participants reported that health facilities use specific criteria to select the members of the PMT, including the head of the quality department, HMIS unit team leader, department head, management team, and staff with good performance.

A respondent stated, "...*since the performance of every department is reviewed in the meeting, the main criteria for PMT membership is being a department head. The heads of management unit, HMIS unit, and quality department are also included. I became a member because I have served as a vice matron for the last one and half years.*" (KI9, 29 years old)

**Table 2. Socio-demographic characteristics of key informants at the health facilities, Eastern Ethiopia 2021.**

| Participant ID | Sex | Age in years | Educational status | Work experience in years | Facility type |
|---|---|---|---|---|---|
| KI1 | Female | 34 | BSc | 11 | Health center |
| KI2 | Female | 25 | Diploma | 2 | Health center |
| KI3 | Male | 27 | BSc | 9 | Health center |
| KI4 | Female | 38 | BSc | 20 | Health center |
| KI5 | Male | 26 | BSc | 7 | Health center |
| KI6 | Male | 34 | BSc | 13 | Health center |
| KI7 | Male | 28 | Master's | 3 | Health center |
| KI8 | Female | 25 | BSc | 1 | Health center |
| KI9 | Female | 29 | BSc | 6 | Referral Hospital |
| KI10 | Male | 28 | BSc | 32 | Referral Hospital |
| KI11 | Male | 32 | BSc | 13 | Health center |
| KI12 | Male | 30 | BSc | 7 | Health center |
| KI13 | Female | 34 | BSc | 8 | Health center |
| KI14 | Female | 39 | BSc | 12 | Health center |
| KI15 | Male | 42 | BSc | 19 | General Hospital |
| KI16 | Male | 35 | BSc | 3 | General Hospital |
| KI17 | Female | 35 | BSc | 12 | General Hospital |
| KI18 | Female | 30 | BSc | 10 | General Hospital |

Another key informant pointed out;

"...any health care provider in a case team can be a member of the PMT. That means there will be one representative from each case team in PMT irrespective of his/her status." (KI15, 42 years old)

**Roles and responsibilities of PMT members** include use of health data to improve the health service performance. The specific roles include preparing action plans, preparing directions to improve service delivery, comparing performance overtime, monitoring monthly performance, providing feedback to various units, building the capacity of health workers, and preparing reports. The present study also found that the PMT members were aware of their responsibilities.

One of the participants stated that "...each member of the PMT is expected to prepare and submit their case team or department reports to the head of the health facility on time. All PMT members will discuss on the submitted report during the monthly PMT meeting in which the reports are crosschecked for its consistency and quality. When the achievements are lower than expected, the reasons for underachievement are sorted out and future directions are put to improve the performance." (KI4, 38 years old)

The participants' opinions on aligning the PMT's roles and responsibilities at the facility with their activities were mixed. While the majority of respondents stated that PMT tasks and responsibilities are aligned with their actual activities, some stated they do not match with their actual activities. There is some overlapping of duties and roles, according to these participants.

"As a case team leader, my responsibility is identifying the challenges in my activities but sometimes the roles and activities rendered to us are not related to service performance and members get frustrated due to that" (KI16, 35 years old)

**Relevance of PMT strategy and its implementation.** **The perceived benefits** of PMT for the facility include improving the service delivery and customer satisfaction. Additionally, this team is essential for sharing experiences and skills among the departments and case teams.

"...the members of PMT are from different departments; there is knowledge sharing on different topics including preparation of reports and how to improve performance." (KI11, 32 years old)

Planning, identifying gaps through root cause analysis, and intervening on the identified problems, and monitoring and evaluating specific activities.

The key informants further pointed out, "...we select poorly performed activities and the responsible departments will design an action plan for the identified problems by conducting a root cause analysis. Then, the department will be directed to monitor the implementation of the action plan and they are expected to bring the progress in the subsequent meetings." (KI17, 35 years old)

**The modality of PMT meetings.** Health facilities have a monthly meeting plan in their health facilities, which is prepared in advance in the annual plan. Respondents pointed out

that the monthly meetings were conducted immediately after each unit submitted their monthly report.

*"Our PMT meeting is conducted on monthly basis. Immediately after the submission of the monthly report (usually at 21st day of the month), we hold our PMT meeting at 23rd or 24th day of the month. Substitution date will be posted on our telegram page if a meeting is rescheduled."* (KI12, 30 years old)

**Adequacy and convenience of timing of PMT meeting.** On the other hand, it was found that the time allocated for PMT meetings was inadequate and inconvenient. All participants unanimously agreed that there should be enough time assigned and a convenient time allocated for the PMT meetings.

*"The meeting time is not enough because every participant usually come to the meeting by interrupting their work. Usually, the meeting date and time is decided by the head."* (KI16, 35 years old)

Participants discussed how holding the meeting on weekends could reduce absenteeism and unnecessary interruptions.

*". . .previously, we used to hold the PMT meeting during working hours and hence the duration of the meetings was not adequate. This was due to high workload of the PMT members. However, currently we are conducting our PMT meeting on Saturdays. Since it is not a working day, PMT members are paid some pocket money for attending the meeting on over the weekend."* (KI8, 25 years old)

**The focus of PMT meetings** reportedly included data quality, performance activities (monthly, quarterly, or annual performance), service quality, and service improvement strategies. According to participants, the first agenda of the PMT meeting was evaluating the previous report, and then comparing it with the current performance. Next, the team develops an action plan based on the identified gaps. However, in some health facilities, the previous performance is not conducted at all.

*"In our monthly PMT meeting, the first thing is presenting the monthly performance report for each unit. Based on the report, gaps and challenges are identified, before action plans are drafted on the identified gaps"*. (KI9, 29 years old)

The PMTs mainly utilize routine data for evaluation and monitoring of service delivery programs. Therefore, there needed to be a culture of using health related data from other sources, such as surveys, assessments, or research findings.

*"In our health center, the main target of PMT is the routine data. The data accuracy and completeness are checked. Service improvement strategies are also discussed."* (KI5, 26 years old)

The respondent further stated *"data collected in our health facility is used for making decisions. I have not seen other data sources from researches or surveys being utilized so far."* (KI5, 26 years old)

## 2. Barriers to data use

**The organizational barriers** to data use at the facility level were poor data quality, being a member of multiple committees and high patient flow, human resource related issues, input related issues, inadequate budget allocation, and lack of performance-based incentives.

*"...PMT members have their own duties in addition to the PMT activities which makes it difficult to collect quality data due to competing priorities. If someone is engaged with overlapping duties, the desired outcome may not be achieved".* (KI3, 27 years old)

Participants reported that they were in at least two other committees in their health facility. This resulted in poor follow-up of the designated PMT activities and time constraint with committee meetings and assignments. According to the participants, multiple committees negatively affected the effectiveness of the PMT activities.

*"...being the member of several committees affects the PMT activities. The aim of including a person in a given team or committee is to perform the task effectively and efficiently in order to bring the intended changes. However, the focus of the individual is usually divided to different committees. In most cases the meeting times is overlapping between different committees"* (KI2, 25 years old)

It was further pointed out:

*"Participation in multiple committees decreases the effectiveness of the PMT. Not only for the PMT, but even the other committees are also adversely affected. I suggest to not assign case team leaders on more than one committee."* (KI14, 39 years old)

The routine data quality collected at the health facilities was reported as a barrier for effective data use.

*"There is a gap in data quality including gaps with data consistency and completeness. Healthcare providers perform their daily activities but they do not document it on the register regularly. The data from HMIS may contradict from your observation every day. Hence it is difficult to use our data for decision making due to its poor quality."* (KI2, 25 years old)

Another participant further pointed out *"A timely, reliable and high-quality data should be generated in order to use the data for decision making. Monitoring and supporting the staffs is also critical to obtain high quality data."* (KI17, 35 years old)

HIS input related factors, including a shortage of patient registrations was cited as another barrier for data use.

*"We have an old register called daily register for documentation of our data. This register was supplied by a Non-Governmental Organization. However, this supply of registers has stopped and currently, we have shortage of the daily registers which affected our documentation."* (KI13, 34 years old)

An additional barrier for effective data use at the health facilities was budget constraints, which was felt to influence documentation in the facilities.

*"We use the budget given for the health center, but there is no budget allocated specifically for HMIS."* (KI6, 34 years old)

**Healthcare providers' work ethic and behavioural barriers** were another sub-theme that emerged in this study. Poor commitment and lack of accountability from healthcare providers and PMT members were the most reported challenges. One respondent explained the commitment and competing priorities of PMT members as

*". . .sometimes we reviewed the same problem repeatedly without a solution. We start to get fed up and start to wonder when it will be solved. Sometimes, we skip the meeting intentionally for this reason.* (KI9, 29 years old)

Another respondent pointed out that:

*". . .health workers that reported incomplete or false data are not given any administrative reprimands for their wrong doing."* (KI18, 30 years old)

Furthermore, a major challenge reported was lack of understanding of data value by the healthcare workers when using data for action.

*"Sometimes the PMT members' attitude towards data is a challenge by itself. The notion that a high-quality data can change the hospital is not understood equally among the staffs."* (KI9, 29 years old)

According to the participants, most of the health professionals have a poor understanding and attitude regarding the importance of data in improving the quality of health services. Additionally, gaps in skills in analyzing data using electronic computing software, such as District Health Information System version 2 (DHIS2) was mentioned.

*"Disparity in understanding the importance of data among our staffs is the other problem. While some people have better understanding towards data, others do not see the value of it."* (KI9, 29 years old)

One respondent further noted:

*"There is a problem on registering the data and handling of a registers. Even after the data is entered into the DHIS2, there is skill gap in analyzing it."* (KI10, 32 years old)

Poor tracking of problems, and a lack of monitoring of action plans is frequently observed according to the key informants.

*"The main problem is that we don't strictly follow the action plans designed in the previous PMT meetings."* (KI2, 25 Years old)

Absenteeism and interruption due to competing priorities were also reported during PMT meetings.

*"As I said earlier, even if it is a monthly meeting, the date and time should be notified ahead of time. Sudden meetings will affect my routine activities and as a result some members of the*

*PMT tend to be absent and some interrupt the meeting to go back to their urgent activities."* (KI13, 34 years old)

Poor documentation practices were another challenge reported by the respondents. First, in order to create a change in the data use culture, the priority should be changing the attitude of the health care workers before other interventions are introduced.

"*Health professionals should believe documenting data primarily benefits patients, themselves and their health facility. Therefore, working on the attitude change towards documentation should be given more emphasis.*" (KI11, 32 years old)

**Demotivating factors** for effective data use practices were cited by the study participants including a shortage of resources, inadequate salary, and inadequate follow-up.

"*Salary is one of the demotivating factors because it does not fit with the job that we under-take.*" (KI10, 28 years old)

"*There is no follow up mechanism after trainings are provided.*" (KI8, 25 years old)

The managers' leadership skills was one of the factors affecting data use in health facilities. Health managers' and case-team/departments' inadequate leadership skills, and lack of value for data were also mentioned as demotivating factors.

"*The head of the facility lacks decisiveness and leadership qualities on this job. I don't think the problem is a gap in knowledge but rather an attitude problem.*" (KI13, 34 years old)

Another respondent pointed out "*. . .if the head of the department or the facility do not uti-lize data properly, other employees will not care about data use. Hence, the head at district, region or at any other level should lead by example to motivate other staffs.*" (KI2, 25 years old)

**COVID-19 effect on data use.**   The COVID-19 pandemic had direct and indirect effects on data use practices at the health facilities. An informant explained the effect of COVID-19 pandemic on their day-to-day activity as

"*Due to the occurrence of COVID -19 outbreak, the nearby hospital was entirely devoted to provide health care service for COVID-19 cases. Hence, all patients in the catchment area of that hospital came to our health center. These conditions increased the workload and compro-mised our routine activities.*" (KI14, 39 years old)

Another key informant stated "*. . .due to COVID-19, the head of HMIS has been in quaran-tine for the last two months which negatively affected the PMT activities.*" (KI3, 27 years old)

## 3. Facilitators of data use

**Motivation mechanism in place for PMT members at the facilities.**   This study identi-fied several motivation mechanisms in place though not across all facilities and not conducted regularly. The main motivation strategies reported were capacity building and performance-based recognition.

**Capacity building practices for PMT members.**   According to participants, the primary capacity building practices included in-service trainings, recognition for good performance, and professional development.

*"Capacity building on data quality and project development for the members of PMT has been a very good incentive since we never had such experience before."* (KI7, 28 years old)

**Recognition and Non-monetary incentive mechanisms** to motivate the PMT members in the facility were reported. Participants indicated that these motivational mechanisms were effective when directed towards improving the data use culture of the facilities.

A participant pointed out that "*performance-based recognition and motivation of staffs can improve the data quality and subsequently the data use in the facility.*" (KI6, 34 years old)

However, in some health facilities there were not any motivation measures specific to PMT members.

*"There are no capacity buildings, recognition mechanisms, and career enhancing educational opportunities to motivate PMT members."* (KI4, 38 years old)

Other participants reported that some sort of recognition and motivation mechanisms were implemented in their facilities.

*"They awarded me this watch for my good performance for properly ensuring the quality of data collected in my health facility."* (KI7, 30 years old)

Others believed that incentives may not necessarily motivate staff to use the data or improve service performance.

*"It is hard to say that the presence of incentives only positively affects data use. Providing incentives before attitude change may even adversely affect the data use practice."* (KI11, 32 years old)

## 4. Recommended mechanisms to improve data use

**Performance-based recognition** for health workers was recommended by all informants as a major motivation mechanism to improve data use.

*"There are those who register and report on time among the health extension workers. The others will be motivated to do the same if you provide recognition or incentives. The incentive can even be in the form of a certificate."* (KI15, 42 years old)

Another participant stated, *"since the PMT members are the backbone of health facilities, designing motivating mechanisms is important."* (KI9, 29 years old)

**Continuous supply of resources**, improving the infrastructures, and strengthening the HIS.

"*Ensuring continuous supply of necessary materials and recognizing best performers should be introduced.*" (KI9, 29 years old)

**Enhancing the quality of routine data** collected was also unanimously believed to enhance data use practice.

*"In order to use data for a decision, first I have to receive the data properly. A timely, reliable and complete data should be collected."* (KI1, 34 years old)

## Discussion

This study explored experiences, facilitators and barriers of the PMT, using the adapted Consolidated Framework for Implementation Research (CFIR), to generate information for decision-making at points of healthcare delivery. The findings revealed that the PMT in most facilities were established according to the standard criteria set by the Ministry of Health (MOH) of Ethiopia [16]. In line with previous studies, our study indicated that the primary responsibility of the PMT was to improve data quality and regularly monitor progress and improve health service performance [15, 16].

Although some irregularities were reported, the monthly meetings were regularly conducted right after each unit submitted their monthly report and before submitting their report to the next level to monitor progress and improve performance. The MOH guideline indicates that the meeting dates, venue and its members should officially be communicated in advance and the meeting should be conducted at least a day ahead of submission of the monthly report to the next level [15, 16, 26].

The present study found the PMT meetings focused on data quality, performance improvement, and evaluation of previous action plans. Previous studies show PMT meetings should start with follow-up activities for gaps highlighted from the previous PMT meeting as the first item on the agenda, followed by an assessment of the progress on those gaps [16, 26]. Meetings and collected data have no value unless action items from meetings are implemented and data are analyzed into meaningful actions [21].

Health facilities should design strategies to minimize the number of committees and integrate similar committees to improve their service provision. Most PMT members in our study were usually involved in at least two other committees in their health facility. This creates poor follow up of the activities set out by the PMT with the committee members being overburdened by meetings and assignments, creating fatigue and a shortage of time to accomplish PMT assignments.

Evidence indicate that data triangulation using various sources, such as original research, community feedback, expert opinions, vital registration, censuses, and routine HMIS data can yield better results [16, 21, 26]. Although there were practice of use of routine internal data, information use from external sources was limited in this study.

Previous studies indicated that an organizational context that supports data collection, availability, and use, the technical aspects of data processes and tools, and the behavior of individuals who produce and/or use data are the main elements of health information use [26, 27]. The major challenges of data use reported in the present study emanate from organizational, behavioral and technical sources including poor data quality, competing priorities, shortage of skilled human power, and lack of performance-based motivation for the health workers.

Healthcare organizations are increasingly required to gather and report data about their performance and respond to the results of consequential quality measurements [28]. Quality data enables healthcare organizations to monitor progress, making decisions for continuous improvement, and provide effective and efficient health services [29]. Our study revealed that although poor quality of data was one of the major challenges for an informed decision making, it has been used for priority setting as well as designing and implementing action plans. This requires attention as decisions taken by healthcare managers might be misleading and fails to address the actual problem. Previous study in Addis Ababa indicated that the PMT meetings that were designed for the sole purpose of improving data quality were not effective [30]. Another qualitative study to explore the facilitators and barriers of digital health technology use also identified data quality as a potential barrier [31]. Moreover, lack of available data for several indicators and a lack of validated indicators for important dimensions of quality

were identified as major challenges to improve Primary Health Care (PHC) Performance in previous studies [6].

Behavioral factors provide crucial insight into the way in which health workers, managers and policymakers use information [26]. The attitude, motivation, and value decision makers attach to information play a big role in determining data use [2, 26]. Concurrent with these, poor commitment, skill gap, lack of accountability for failing to accomplish tasks, work overload, poor documentation, and resource mismanagement were the main behavioural level challenges reported in this study.

Engagement with different committees increased PMT workload and compromised the effectiveness of data use. Participation in quality improvement collaborative activities may improve health professionals' knowledge, problem-solving skills and attitude; teamwork; shared leadership and habits for improvement. A review of literature indicated shifting roles and expectations in the workplace for health care leadership and management as a major challenge for health leadership and workforce [1]. Interaction across quality improvement teams may generate normative pressure and opportunities for capacity building and peer recognition [15].

Health professionals either do not consider recording and reporting data as part of their routine activities or they just give more priority to the clinical provision and lesser attention to data [30]. Although there is a need to make significant investments in the workforce development and training [32], tackling of behavioral factors require interventions that go beyond simple trainings and should focus more on initiatives that enhances positive attitude towards data value [26].

The individual determinants of motivation among healthcare workers are altruism, attaining professionalism, educational opportunity, and being more experienced [33]. In the present study, it was indicated that health facilities use capacity buildings, and some form of non-monetary incentives to improve performance, while most informants indicated dissatisfaction with the absence of such motivating mechanisms. Career development opportunities, in-service trainings, and regular recognition for good performance were also reported to be practiced at the health facilities which has to be expanded to more institutions.

Provision of inputs necessary for HIS such as laptops, internet modems; as well as awarding the best performing health facilities were other motivational methods. Since health professionals can be motivated by a range of extrinsic and intrinsic factors, policy makers need to look beyond traditional financial incentives when designing policies to improve health service performance [34]. Studies indicate training opportunities, transformative leaders, and nonmonetary incentives, staff development and promotion, availability of necessary resources, and ease of communication are found to be effective motivation mechanisms for health care workers [33, 35].

## Conclusions

This study has generated important insights into experiences of PMT including its establishment, and implementation strategies, barriers to data use, and facilitators of using information for decision at a point of service delivery. The study found that most performance monitoring teams in health facilities were established and functioning according to the national standard. Additionally, the study revealed that the barriers to effective data use include the PMT attending multiple committee meetings (increasing workload), poor data quality, lack of accountability, and poor documentation practices. Recommendations to enhance data use practices include non-monetary incentives and recognition for exemplary health workers. Improving routine data quality, integrating various teams into PMT for an efficient use of the limited

human resource, establishing an accountability framework, and designing documentation methods will ultimately improve informed decision-making. While a comprehensive country-wide study of PMTs is required, policymakers, stakeholders working on HIS, and health managers should work on improving routine data quality and design motivational strategies, including recognition and non-monetary incentives to improve data use which has the potential to improve health service performance.

## Acknowledgments

The authors are thankful to the Ethiopian Ministry of Health, Dire Dawa and Harari Regional Health Bureaus. We are also grateful for participating health facilities and their healthcare staffs for their time and voluntarily for contributing to the research. Moreover, we would like to acknowledge our data collectors and logistic facilitators for their support throughout the study.

## Author Contributions

**Conceptualization:** Admas Abera, Tilahun Shiferaw, Alemayehu Girma, Rania Mohammed, Yadeta Dessie.

**Data curation:** Admas Abera, Biruk Shalmeno Tusa, Adisu Birhanu Weldesenbet, Assefa Tola, Alemayehu Girma, Rania Mohammed.

**Formal analysis:** Abebe Tolera, Biruk Shalmeno Tusa, Adisu Birhanu Weldesenbet, Assefa Tola, Tilahun Shiferaw.

**Funding acquisition:** Admas Abera, Rania Mohammed.

**Investigation:** Admas Abera, Adisu Birhanu Weldesenbet, Alemayehu Girma.

**Methodology:** Admas Abera, Abebe Tolera, Biruk Shalmeno Tusa, Assefa Tola, Tilahun Shiferaw, Alemayehu Girma, Rania Mohammed, Yadeta Dessie.

**Project administration:** Admas Abera.

**Software:** Abebe Tolera, Assefa Tola.

**Supervision:** Admas Abera, Adisu Birhanu Weldesenbet, Alemayehu Girma, Rania Mohammed, Yadeta Dessie.

**Validation:** Admas Abera, Abebe Tolera.

**Visualization:** Admas Abera.

**Writing – original draft:** Admas Abera, Abebe Tolera, Assefa Tola.

**Writing – review & editing:** Admas Abera, Biruk Shalmeno Tusa, Tilahun Shiferaw, Alemayehu Girma, Rania Mohammed, Yadeta Dessie.

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
