## [Decision Letter · Decision Letter 0]

16 Aug 2022

PONE-D-22-10622Experience, Challenges, and Facilitators of Health Data Use among Performance Monitoring Teams (PMT) of Health Facilities in Eastern Ethiopia: A Qualitative Study.PLOS ONE

Dear Dr. Abera,

Thank you for submitting your manuscript to PLOS ONE. After careful consideration, we feel that it has merit but does not fully meet PLOS ONE’s publication criteria as it currently stands. Therefore, we invite you to submit a revised version of the manuscript that addresses the points raised during the review process.

Please note that we have only been able to secure a single reviewer to assess your manuscript. We are issuing a decision on your manuscript at this point to prevent further delays in the evaluation of your manuscript. Please be aware that the editor who handles your revised manuscript might find it necessary to invite additional reviewers to assess this work once the revised manuscript is submitted. However, we will aim to proceed on the basis of this single review if possible.

The reviewers have raised a number of concerns that need attention. They request additional information on methodological aspects of the study, revisions to the statistical analyses and they question the internal and external validity of the results reported.

Could you please revise the manuscript to carefully address the concerns raised?

We look forward to receiving your revised manuscript.

Kind regards,

Thomas Phillips, PhD

Staff Editor

PLOS ONE

Journal Requirements:

Reviewers' comments:

Reviewer's Responses to Questions

**Comments to the Author**

1. Is the manuscript technically sound, and do the data support the conclusions?

Reviewer #1: Yes

2. Has the statistical analysis been performed appropriately and rigorously? 

Reviewer #1: Yes

3. Have the authors made all data underlying the findings in their manuscript fully available?

Reviewer #1: Yes

4. Is the manuscript presented in an intelligible fashion and written in standard English?

Reviewer #1: Yes

5. Review Comments to the Author

Reviewer #1: Dear Editor

I trust that this finds you in good spirits.

I appreciate the opportunity to review the manuscript and have found pleasure in studying it through.

Use of data in health facilities is an important factor that guides operations to where they matter most and further helps in planning for both short term and long term.

This manuscript have identified several factors contributing to both use and failure to use data as purported.

The following require attention for the manuscript to improve:

Formatting: Attention needs to be paid on following on prescribed formatting rules like how to write titles, citation and referencing, etc.

Abstract:

-few typo suggestions

Introduction:

-Typo suggestions and several referencing style inconsistency Generally, the manuscript does not satisfactorily follow Vancouver style as prescribed in guidelines.

Methods:

-Typing and language recommendations have been made.

-Under data analysis, authors need to consider matters of backward translation.

-This section mentions that there were key themes and there is no mention of sub-themes yet there seem to be several of these Also indication on how trustworthiness was ensured is important.

Results:

-Emerged themes require regrouping/renaming/restructuring.

-Results Data under the themes can be better understood if themes are framed differently.

-A fifth theme on relevance of PMT strategy and its implementation has been recommended as some of subthemes do not fit well in current themes

-Theme 1 can be re-titled to 'Membership, roles and experiences of PMT members at the health facilities'

-Given the sample size for the study, and generally qualitative studies, study identifiers in this study are likely to compromise anonymity. Please consider using other more neutral identifier like participants ID. With other facilities, it happens that there is only one program planning team leader and subsequently someone reading the manuscript could easily identify who said what.

-Authors have often discussed participants' responses under results as opposed to showing statements from participants under results then discuss under discussions. Subsequently, some participants' statement are missing under results.

-Transcriptions require to be confirmed as one of the statements is ambiguous and may lead the reader to confusion (Since it is not a working day, we are paying some pocket money for the PMT members.”)

-'Effect of multiple committees in the facility on PMT' subtheme can be better positioned under Barriers

Discussion:

-'On the other hand, health workers reported the desire to engage in on-the-job learning, workshops,

seminars, refresher courses and continuous training.' This sentence dose not reflect what transpired under results.

-

6. PLOS authors have the option to publish the peer review history of their article (what does this mean?). If published, this will include your full peer review and any attached files.

Reviewer #1: **Yes: **Sibusiso Nomatshila

---

## [Author Response · Author response to Decision Letter 0]

26 Aug 2022

All the comments raised by the reviewer have been addressed and attached in a point-by-point response in the attached "response to Reviewers" word document. Kindly refer the responses on the document as they are detail and long. Thank you.

---

## [Decision Letter · Decision Letter 1]

15 Feb 2023

PONE-D-22-10622R1Experience, Barriers, and Facilitators of Health Data Use among Performance Monitoring Teams (PMT) of Health Facilities in Eastern Ethiopia: A Qualitative Study.PLOS ONE

Dear Dr. Abera,

Thank you for submitting your manuscript to PLOS ONE. After careful consideration, we feel that it has merit but does not fully meet PLOS ONE’s publication criteria as it currently stands. Therefore, we invite you to submit a revised version of the manuscript that addresses the points raised during the review process.

ACADEMIC EDITOR: Based on the reviewer comments and my own observations I will recommend Minor Revisions on the paper. Please pay attention to the comment provided by reviewer 2 and below points:Make sure that the formatting of the manuscript is according to the journal’s requirements. Include required sections following the conclusions statement. Use square brackets (i.e., [   ]), rather than (   ) for inside text citation of references. Full stops should come after the in-text citations (i.e., [1,3]. )Briefly explain how the CFIR framework has been applied to guide the present study. The organization of the methods sections has unclarity. The description of sample size and sampling strategy should be provided prior to data collection procedures. Thus, it is highly recommended if it is restructured as follows:  Study settingStudy design and participants: here try to briefly describe the study design, sample size, sampling strategy, and inclusion/exclusion criteria. Data collectionData quality control: here briefly explain the strategies used to ensure the quality of data. Data processing and analysisEthical consideration 
There are grammar and language flaws that requires revisions.  For instance: take a look at the statement in page 11 that starts with: 'planning, identifying gaps and intervening****'; page 12: 'the focus of PMT meeting****etc..."The paper has reported a very good results, however the discussion is weaker. More evidences/literatures should be used to support the justifications and the implications of observed results should be briefly presented. Moreover, authors are highly suggested to discuss the limitations of the present study (if any) 

A rebuttal letter that responds to each point raised by the academic editor and reviewer(s). You should upload this letter as a separate file labeled 'Response to Reviewers'.A marked-up copy of your manuscript that highlights changes made to the original version. You should upload this as a separate file labeled 'Revised Manuscript with Track Changes'.An unmarked version of your revised paper without tracked changes. You should upload this as a separate file labeled 'Manuscript'.We look forward to receiving your revised manuscript.

Kind regards,

Dawit Wolde Daka

Academic Editor

PLOS ONE

Reviewers' comments:

Reviewer's Responses to Questions

**Comments to the Author**

1. If the authors have adequately addressed your comments raised in a previous round of review and you feel that this manuscript is now acceptable for publication, you may indicate that here to bypass the “Comments to the Author” section, enter your conflict of interest statement in the “Confidential to Editor” section, and submit your "Accept" recommendation.

Reviewer #1: All comments have been addressed

Reviewer #2: (No Response)

2. Is the manuscript technically sound, and do the data support the conclusions?

Reviewer #1: Yes

Reviewer #2: Yes

3. Has the statistical analysis been performed appropriately and rigorously? 

Reviewer #1: Yes

Reviewer #2: Yes

4. Have the authors made all data underlying the findings in their manuscript fully available?

Reviewer #1: Yes

Reviewer #2: (No Response)

5. Is the manuscript presented in an intelligible fashion and written in standard English?

Reviewer #1: Yes

Reviewer #2: Yes

6. Review Comments to the Author

Reviewer #1: (No Response)

Reviewer #2: The paper has scientific merit provided that the authors include or justify the given comments. Most importantly, how they utilized CFIR framework in presenting their results. In addition, in the discussion section, the authors should give clear justification on how the PMT has been used the poor data (reported as a barrier) for performance improvements.

7. PLOS authors have the option to publish the peer review history of their article (what does this mean?). If published, this will include your full peer review and any attached files.

Reviewer #1: **Yes: **SC NOMATSHILA

Reviewer #2: No

---

## [Author Response · Author response to Decision Letter 1]

24 Mar 2023

The comments and questions raised by both the academic editor and reviewer were insightful and helped improve our manuscript. We have gone through each comments raised and attached under the authors responses in the system. We would like to express our deep gratitude. Thank you!

---

## [Decision Letter · Decision Letter 2]

18 Apr 2023

PONE-D-22-10622R2Experiences, Barriers, and Facilitators of Health Data Use among Performance Monitoring Teams (PMT) of Health Facilities in Eastern Ethiopia: A Qualitative Study.PLOS ONE

Dear Dr. Abera,

Thank you for submitting your revised manuscript to PLOS ONE. Though majority of the previously provided comments are addressed, still there are few issues from reviewer 2 that requires revisions and responses. Providing responses only may not be sufficient and the two issues raised by reviewer 2 should be addressed within the manuscript to make clearer to readers. Thus, pay attention to those comments and submit the revised version of the manuscript. 1. Within the manuscript briefly describe the themes and sub-themes related to experience, barriers and facilitators2. In the methods, provide more information regarding how CFIR framework guided the present research and at what stage of the research. Reviewer comments are indicated in the attached file. Please submit your revised manuscript by Jun 02 2023 11:59PM. If you will need more time than this to complete your revisions, please reply to this message or contact the journal office at plosone@plos.org. Please include the following items when submitting your revised manuscript:A rebuttal letter that responds to each point raised by the academic editor and reviewer(s). You should upload this letter as a separate file labeled 'Response to Reviewers'.A marked-up copy of your manuscript that highlights changes made to the original version. You should upload this as a separate file labeled 'Revised Manuscript with Track Changes'.An unmarked version of your revised paper without tracked changes. You should upload this as a separate file labeled 'Manuscript'.We look forward to receiving your revised manuscript.

Kind regards,

Dawit Wolde Daka

Academic Editor

PLOS ONE

Reviewers' comments:

Reviewer's Responses to Questions

**Comments to the Author**

1. If the authors have adequately addressed your comments raised in a previous round of review and you feel that this manuscript is now acceptable for publication, you may indicate that here to bypass the “Comments to the Author” section, enter your conflict of interest statement in the “Confidential to Editor” section, and submit your "Accept" recommendation.

Reviewer #2: (No Response)

2. Is the manuscript technically sound, and do the data support the conclusions?

Reviewer #2: Yes

3. Has the statistical analysis been performed appropriately and rigorously? 

Reviewer #2: Yes

4. Have the authors made all data underlying the findings in their manuscript fully available?

Reviewer #2: (No Response)

5. Is the manuscript presented in an intelligible fashion and written in standard English?

Reviewer #2: Yes

6. Review Comments to the Author

Reviewer #2: I am grateful to the authors for submitting the revised manuscript. Their hard work and dedication is impressive.

However, I noticed that some of my comments in the result section were not addressed in their revised manuscript.

Please also see the attached doc.

7. PLOS authors have the option to publish the peer review history of their article (what does this mean?). If published, this will include your full peer review and any attached files.

Reviewer #2: No

---

## [Author Response · Author response to Decision Letter 2]

26 Apr 2023

Dear Editor and Reviewers,

Thank you for your comment on our manuscript. We have tried to revise all the remaining comments on the manuscript. We hope we have addressed your concerns in this round. Thank you.

---

## [Editor Report · Decision Letter 3]

28 Apr 2023

Experiences, Barriers, and Facilitators of Health Data Use among Performance Monitoring Teams (PMT) of Health Facilities in Eastern Ethiopia: A Qualitative Study.

PONE-D-22-10622R3

Dear Dr. Abera,

We’re pleased to inform you that your manuscript has been judged scientifically suitable for publication and will be formally accepted for publication once it meets all outstanding technical requirements.

Kind regards,

Dawit Wolde Daka

Academic Editor

PLOS ONE
---

## [Editor Report · Acceptance letter]

3 May 2023

PONE-D-22-10622R3 

Experiences, Barriers, and Facilitators of Health Data Use among Performance Monitoring Teams (PMT) of Health Facilities in Eastern Ethiopia: A Qualitative Study. 

Dear Dr. Abera:

I'm pleased to inform you that your manuscript has been deemed suitable for publication in PLOS ONE. Congratulations! Your manuscript is now with our production department. 

Kind regards, 

on behalf of

Mr Dawit Wolde Daka 

Academic Editor

PLOS ONE